# In Vitro Effects of Cyclic Dislodgement on Retentive Properties of Various Titanium-Based Dental Implant Overdentures Attachment System

**DOI:** 10.3390/ma12223770

**Published:** 2019-11-16

**Authors:** Tae-Yun Kang, Jee-Hwan Kim, Kwang-Mahn Kim, Jae-Sung Kwon

**Affiliations:** 1Department and Research Institute of Dental Biomaterials and Bioengineering, Yonsei University College of Dentistry, Seoul 03722, Korea; tykang@yuhs.ac (T.-Y.K.); kmkim@yuhs.ac (K.-M.K.); 2BK21 PLUS Project, Yonsei University College of Dentistry, Seoul 03722, Korea; 3Department of Prosthodontic, Yonsei University College of Dentistry, Seoul 03722, Korea; jee917@yuhs.ac

**Keywords:** titanium, dentures, dental implants, denture precision attachment, prosthesis retention, reference standard

## Abstract

The purpose of this study was to evaluate the change in the retentive forces of four different titanium-based implant attachment systems during the simulation of insert–removal cycles in an artificial oral environment. Five types of titanium-based dental implant attachment systems (Locator, Kerator, O-ring, EZ-Lock, and Magnetic) were studied (n = 10). The specimens underwent insert–removal cycles in artificial saliva, and the retentive force was measured following 0, 750, 1500, and 2250 cycles. Significant retention loss was observed in all attachment systems, except the magnetic attachments, upon completion of 2250 insertion and removal cycles, compared to the initial retentive force (*p* < 0.05). A comparison of the initial retentive forces revealed the highest value for Locator, followed by the Kerator, O-ring, EZ-Lock, and Magnetic attachments. Furthermore, Kerator demonstrated the highest retentive loss, followed by Locator, O-ring, EZ-Lock, and Magnetic attachments after 2250 cycles (*p* < 0.05). In addition, the Locator and Kerator systems revealed significant decrease in retentive forces at all measurement points (*p* < 0.05). The retention force according to the insert–removal cycles were significantly different according to the types of dental implant attachment systems.

## 1. Introduction

Titanium-based implant-supported overdentures (IOD) have been used as an alternative to conventional dentures for the oral rehabilitation of edentulous patients [1,2]. Titanium implants sustain the residual alveolar bone, thereby, significantly improving the retention, esthetics function, and stability of dentures [3,4]. A previous study reported that IOD improves the chewing efficiency and maximum occlusal force of patients, compared to conventional complete dentures; thus, it significantly improves edentulous patients’ quality of life [5,6,7].

IODs can be provided with various types of attachments, including ball, locator, and magnet-type attachments on implants. Currently, there are several types of each attachment system in the market varying in shape, materials, and retention properties [8,9,10,11]. 

Success of the treatment plan in the edentulous state has been measured mainly based on the patient’s opinion [10]. As the retention force of the attachments systems is related to patients’ satisfaction with implant overdentures and associated with the stability and adequate retention force of the denture, it has been the primary consideration when clinicians have selected different types of attachments [5,12,13,14]. 

The most common problem associated with attachment systems is the loss in retention caused by wear, deformation, and fracture of the components of the attachment system over time, which is related to repeated insertion and removal of the attachment components by the cyclic dislodgement of attachments [15,16]. For the safe application of dental implant attachments in a patient’s oral environment, general information about the product, such as the initial retention of the product and the tendency of retention loss, must be provided. However, a very limited amount of data is available on dental implant attachments pertaining to the retentive force of the attachment and the retentive loss arising from cyclic insertion and removal [7,10]. Therefore, previous investigations have attempted to characterize the retention of attachments used for IOD and compare the retention force characteristics of various implant attachment systems [5,6,7,17]. However, these studies were conducted under varying experimental conditions; even for the same attachment system, differences of cross head speed, number of implants, and cyclic loading and implantation angles exist [6,7,10]. Consequently, it is difficult to objectively compare the retention characteristics of the attachment systems in relation to the effect of cyclic insertion and removal of attachment system using the previous research results [16,18,19,20,21,22]. 

Therefore, the objective of this study was to compare the changes in the retentive force of the five types of commonly used attachment systems following the simulation of insertion–removal cycles in an artificial oral environment.

The null hypotheses of this study were as follows: (1)There would be no significant differences in the retention force of the attachment systems before and after undergoing the cyclic insertion–removal procedures.(2)There would be no significant differences in the retention force between the five different attachment systems.

## 2. Materials and Methods

### 2.1. Materials and Preparation of the Specimens

The five types of titanium-based dental implant attachment systems used in this study are shown in Table 1. To simulate the clinical use of implant attachment overdentures, each attachment specimen was embedded in a custom-manufactured titanium-alloy jig with autopolymerizing acrylic resin (GC Pattern Resin, Tokyo, Japan), as shown in Figure 1. All attachment abutments were tightened on the implant’s fixture (TSⅢ SA, 5.0 mm × 1.5 mm, OSSTEM IMPLANT Co., Seoul, Korea) with a manufacturer’s recommended torque value using a ratchet and torque control device (Figure 2.). For each group, 10 specimens were used (n = 10). 

### 2.2. Retention Force Measurement of Specimens

The initial retentive force of the specimen was measured based on an adaptation of the International Standard, ISO 13017:2012 Dentistry—Magnetic attachments. Briefly, a universal testing machine (UTM, Instron 5942, Intron, Norwood, MA, USA) with crosshead speeds of 5 mm/min was used. 

The specimens were evaluated based on loss of retentive force after 750, 1500, and 2250 cycles of insertion and removal. Interimplant divergence angles of 0 degrees were selected. Based on the assumption that under regular circumstances, a patient places/removes their implant overdenture four cycles/day (after every meal and before sleep time), the attachments included in this study were tested for loss of retention after being subjected to 750 (6 months), 1500 (12 months), and 2250 (18 months) cycles of insertion–removal simulation [23,24,25]. The retentive forces were calculated from five measurements at each interval using the universal testing machine (UTM) and Bluehill^®^ 2 software (Intron) for each specimen. The lower structure of the specimens was attached to a custom-made water chamber and seated on the lower jig of the UTM. Furthermore, through the tensile test jig, the upper structure of the specimens was attached to the upper jig. 

To simulate the wet oral environment, the custom-made detachable water chamber was filled with artificial saliva. All specimens were subjected to insertion–removal cyclic loading in a simulated wet oral environment at a cyclic rate of 20 cycles/min, using standard artificial saliva (AS). The AS was prepared in accordance with the International Standard, ISO 10271:2011 Dentistry—Corrosion test methods for metallic materials. The testing solution and specimens were kept at (37 ± 2) °C during testing, in conformity with the previous studies [26,27,28]. The mean retentive force and standard deviations were calculated from five repeated measurements for each specimen. The percentage of retention loss was calculated using the following equation:$$Retention\;loss\;\left(\%\right)=\;\frac{\mathrm{Initial}\;\mathrm{retentive}\;\mathrm{force}-\;\mathrm{final}\;\mathrm{retentive}\;\mathrm{force}\;}{\mathrm{Initial}\;\mathrm{retentive}\;\mathrm{force}}\times 100.$$

### 2.3. Surface Morphology

After the insertion–removal cyclic test, one specimen from each group was randomly selected and analyzed through scanning electron microscopy (SEM; 7800, JEOL, Tokyo, Japan). The SEM images were evaluated to determine possible wear patterns on the patrix and matrix surface during the cyclic tests.

### 2.4. Statistical Analysis

For the statistical analysis of the results, SPSS (IBM, Armonk, NY, USA) was used. The one-way analysis of variance (ANOVA) statistical analysis was applied with Tukey’s test for post-hoc analysis. In all the tests, the level of significance was set to *p* < 0.05.

## 3. Results

### 3.1. Retention Force of Specimens

Figure 3 shows the mean values of the retentive force calculated in this study. Furthermore, the mean retention force values of the attachment systems at each cycle and the significance of the retention loss in comparison to the previous measurement cycles are presented in Table 2.

The initial retentive forces of the locater attachments (both LOC and KER) significantly exceeded those of the other attachments tested (*p* < 0.05). The highest initial retentive force was demonstrated by the LOC, followed by the KER, EZL, ORI, and MAG attachments. The highest final retentive force after insertion and removal cycles was demonstrated by the LOC, KER, EZL, ORI, and MAG attachments, in that order. Statistical analysis revealed that the LOC, KER, and ORI groups demonstrated significant variations in retention loss with an increasing number of cycles (*p* < 0.05). The highest retention loss rate was observed in the KER, followed by the LOC, ORI, EZL, and MAG attachments, in that order. The KER attachments exhibited a rapid loss of approximately 53.06% of initial forces after the 2250th cycle. For same cycles, LOC, ORI, and EZI exhibited 38.98%, 21.09%, and 18.73% loss, respectively. The MAG attachment, by contrast, exhibited only 3.38% loss. 

The MAG attachments exhibited the least retention force and retention loss. No significant change was noted over time (*p* > 0.05). Although the initial retentive forces of the locater and KER attachments significantly exceeded that of the other attachments, it remarkably decreased after 2250 cycles (*p* < 0.05). A significant decrease was noted in the retentive force of the ORI at all measurement steps (*p* < 0.05). There were no significant differences in the retentive forces of the MAG attachments before and after cyclic loading (*p* > 0.05). This system remained stable through all the cycles. The initial retentive force of the EZL attachments was significant different when the force was measured after 750 cycles (*p* < 0.05), but no significant differences were observed between 750 and 2250 cycles *p* > 0.05). Additionally, Table 2 shows the result of the retention force among the attachment systems after the same number of cyclic insertion and removal of attachments are performed at each interval. The retention force of LOC and KER is significantly different at all five measurement intervals, compared to the other groups (*p* < 0.05). At the beginning, and in the 750th and 1500th cycle, there were no significant differences of the retention force between the ORI and EZL groups (*p* > 0.05). In the 2250th cycle, there were no significant variations in the retention forces among the MAG, ORI, and EZL groups (*p* > 0.05). Finally, MAG exhibited the lowest retentive values at all the measurement points, except for the final retention force (after 2250 cycles).

### 3.2. Surface Morphology

The SEM images showed the morphological appearance of the five attachment systems before and after the cyclic insertion–removal (after 2250 cycles) (Figure 4). There were slightly more scratches on EZL (Figure 4C) compared to the others. With regard to the attachment replacement for all the attachment systems, the ORI (Figure 4F), LOC (Figure 4H), and KER (Figure 4I) replacements were found to have wear and deformation on the internal and external surfaces, while the EZL (Figure 4G) replacement did not exhibit any perceivable scratches or deformation. 

## 4. Discussion

The use of titanium-based implant to treat edentulous patient has become a well-established clinical treatment method [29]. Still, the results of the previous studies have shown that clinical and biomechanical aspects such as implant body design and implant surface treatment would influence the success of clinical outcomes [30,31,32,33]. Therefore, in order to achieve successful clinical outcomes, both clinical and biomechanical aspects would need to be considered in relation to the device. This study considered different attachment systems in relation to retention force that would be influenced by insertion–removal cycles.

The present in vitro study investigated the retentive force of five implant attachment systems during the simulation of insertion–removal cycles in an artificial oral environment. According to the results of this study, the first null hypothesis that the number of insertion–removal cycles does not affect the retention force of the attachment specimens was partially accepted. Although there were significant differences in the retentive force of LOC, KER, and ORI as the number of the insertion–removal cycles increased, there were no significant changes in the retention force value of the MAG and EZL groups (The MAG group exhibited no significant decrease in retention force at all measurement points. The EZL group exhibited no significant decrease in retention force after 750, 1500, and 2250 cycles). The second null hypothesis, that there would be no difference in the retention force of the five attachment systems after the same number of insertion and removal cyclic loading, was also partially accepted. Despite differences in the retentive force of LOC and KER, there were no significant differences between the MAG, EZL, and ORI groups.

Although implant attachment systems should have sufficient retentive force to prevent prostheses from falling out of the mouth, products with a large retention force are prone to make patients feel discomfort during insertion and removal. In this respect, information on the “period of retention release” of the attachment systems is important. The period of retention release may be described as the time required for the attachment system to lose retention force to support the IOD. This information has obvious clinical significance for retention and stability of the IOD during function in patient’s oral care [34].

An appropriate attachment release period may serve as a mechanical safety measure unique to that attachment system [11,20,35]. Furthermore, the retentive force value is very important, and is the most fundamental consideration in the selection of attachment systems clinically [23,36,37]. Previous studies reported that the longevity of attachment systems are affected by factors such as the number and position of the implants, type and material of the attachments, and design of the prosthesis [11,22,23,37,38,39,40,41], hence, the reason why different types and materials of attachments were considered in this study.

In this present study, we used the AS in order to simulate the oral environment and for the data that is clinically relevant [27]. The use of a lubricant that simulates clinical conditions is absolutely necessary to simulate wear or fatigue because the retention force changes are hugely influenced by these specific conditions [18]. Furthermore, in this study, the AS and the test specimens are kept at (37 ± 2) °C during testing. A previous study investigated the changes in locator-type attachment systems after exposure to different temperatures of distilled water (DW) (20, 37, and 60 °C) and cycling loading; there was a significantly higher rate of retention loss in comparison to specimens immersed in lower-temperature DW [42]. The specimens subjected to high-temperature solution after cyclic insertion–removal loading exhibited significant changes in surface characteristics and cracked the central and outer wall of the locator attachment replacements. This was associated with the hydrolytic degradation of nylon at high temperature. The degradation of the nylon used in locator attachments would lead to a significant loss of retention force during repeated insertion and removal of the implant attachment system [42].

The limitation of this study would be the test that has been carried out on one simulation temperature of (37 ± 2) °C rather than specimen undergoing thermocycling of cold and hot temperature. The thermocycling test simulates the hot and cold environment in the oral cavity that occurs through eating and drinking [43]. A previous study showed the changes in 4 magnetic-type attachment systems after exposure to 10,000 thermal cycles of DW water between 5 and 55 °C, and the results revealed that such thermocycling caused a significant decrease in retentive force in open-field-type magnetic attachments, when compared with their initial retentive force [43]. In addition, the decrease in retention force of the LOC, ORI, and bar-clip-type attachments after subjecting them to 5000 thermal cycles resulted in statistically significant decrease in attachment systems in another previous study [44]. As the information would be available with thermocycling in the previous studies, and as the study design of having thermocycling set-up at the same time of having repeated insertion–removal cycles would be difficult, this study considered a technical environment with lubricant (saliva) temperature of 37 °C rather than considering the effect of thermal variation of intraoral condition.

The retention force of the specimens was measured using a crosshead speed of 5 mm/min, in accordance with the ISO 13017, in this study. Previous studies measured the retentive force of each attachment system at various crosshead speeds, including 50, 50.8, and 60 mm/min [11,21,45]. These crosshead speeds have been reported to approximate clinically relevant movement speed of denture removal from the edentulous ridge [11] as well as the falling movement speed in the mouth during mastication [46]. However, patients may remove their dentures at different speeds, which in turn will affect the retention force [19,47]. In a study where the retentive forces of three attachment types were measured using either slow (2 mm/min) or fast (50 mm/min) crosshead speeds [46], it was evident that the fast speed results in larger retentive force value, compared to the values at slow speed in most of the cases, excluding the magnetic types that demonstrated less retention value at fast speed [48]. Furthermore, there were significant differences between the values of the retention forces at slow (0.5 mm/min) and fast crosshead speeds (50 mm/min) in magnetic systems [49]. Testing the magnetic attachment system at fast crosshead speed revealed a decrease in the retentive force, compared to the slow crosshead speed, and has been indicated to be inaccurate [48,49]. Hence, the low crosshead speed of 5 mm/min has been used in this study. The LOC and KER specimens are recognized for their self-aligned properties and vertical resiliency [16]. Furthermore, these attachments are commercially available in various colors with different retention values [4,48]. In this experiment, the blue color KER patrix and transparent color LOC patrix were used. Thus, the two attachment systems are the same type and material but for the LOC, the evaluation of the retention force value showed a retention value higher than the KER. The retentive values of these systems depend on the patrix, comprising of a metallic cap with replaceable polymer elements. Therefore, these systems would allow easy adjustment of the retentive force [39,49]. However, the matrix of both systems consist of synthetic polymer; therefore, the deviation of the mean values of the retentive force and the retention loss were distinctly marked [16,41], as evident from the results for LOC and KER shown in Figure 3. As demonstrated by the SEM images shown in Figure 4, LOC and KER exhibited deformation and wear behaviors. The wear of the LOC and KER attachments polymer replacements was more rapid, compared to other systems. Previous clinical studies reported that LOC attachments can last up to 1.8 years [19].

The ORI attachment system replacement consists of polymeric materials. Therefore, a significant decrease was noted in the retentive force of the ORI at all measurement points. In addition, the retention loss exhibited by the ORI attachments after 2250 cycles was approximately 21.09% of the initial forces, which may indicate that this system was relatively susceptible to environmental factors, such as friction and heat. It is recommended that the ORI attachment be positioned perpendicularly to the occlusal plane. A lack of parallelism will interfere with the insertion and removal of the prosthesis, leading to an unacceptable wear of the matrix structure [50,51]; the periodic replacement of ORI generally lasts from six to nine months [52].

The EZL system consists of titanium alloy rings with three-unit zirconia balls in the metal housing. EZL is a solitary, resilient type of attachment system [16,22]. EZL exhibited slight retention loss after repeated insertion and removal, and approximately 18.73% of the initial force after the 2250 cycles. As demonstrated by the SEM images taken in this study (Figure 4), wear and deformation were markedly evident on metal abutment. On the other hand, no scratch was observed on the EZL replacement surface, probably due to the zirconia ball and titanium alloy ring present in the EZL attachments [16].

The retentive force of the MAG attachment systems is caused by attractive forces between the N pole and S pole or the pulling forces between the magnetic structure and keeper [53]. These systems are recommended because they are easier to manage than other systems. Furthermore, due to additional advantages including cost-effectiveness, simple fabrication design, and easy repair process, the system has been popularly utilized [54]. Nevertheless, these attachments systems provide the least retention force value compared to any other attachments [35,52]. This system positions itself automatically when it is in proximity to the correct seat position, a particularly useful property for patients with limited manual dexterity and their caregivers [38]. In this study, the MAG attachments demonstrated the weakest retentive force value, but the least change in retentive force value. As demonstrated by the SEM images taken in this study (Figure 4A), slight scratching of surface was observed with the MAG abutment.

This study investigated the retention characteristics of five implant attachment systems. Each system exhibited different retention characteristics. Currently, no guideline exists for the selection of attachment systems, with regard to the retention property when used as IOD. Therefore, the dental clinician must consider the characteristics and information of each attachment system in relation to concerning indications when applied to the patient’s oral cavity.

This study has several limitations. This in vitro study could not sufficiently simulate the movement of the IOD, in relation to a complex and variable biomechanical environment, such as occlusal and mastication forces. Additionally, this study did not simulate the thermal variations of oral cavity due to the difficulty in test method set-up, as discussed earlier. Furthermore, the results of the present study apply only to overdentures retained with single implant. However, the use of several numbers of implants increases the potential for variations in the different retention characteristics corresponding to variations in alignment and position [19]. These factors likely significantly influence attachment retention. Therefore, these results should be interpreted with caution, as the differences in retention value may not be significantly different clinically.

## 5. Conclusions

Within the limitations of this in vitro study, the first null hypothesis was partially accepted. The ORI, EZL, LOC, and KER attachment systems, excluding the MAG attachment (*p* > 0.05), exhibited a significant decrease in retention force (*p* < 0.05) after 2250 cycles. The retentive force and retention loss of the MAG attachments is markedly lower compared to the other attachment systems (*p* < 0.05). The second null hypothesis was partially rejected, because the LOC and KER groups revealed significant difference in retention force at all measurement points (*p* < 0.05) when compared to the other groups. Finally, the LOC attachment exhibited the highest retention of all the attachment systems; however, retention loss was markedly evident (*p* < 0.05). Furthermore, the KER and ORI attachment system revealed a significant decrease in the retention force at all measurement points (*p* < 0.05). The wear of these systems’ polymer components occurs more quickly. However, the EZL attachment exhibited no significant differences after 750 and 2250 cyclic loading (*p* < 0.05).

## Figures and Tables

**Figure 1 materials-12-03770-f001:**
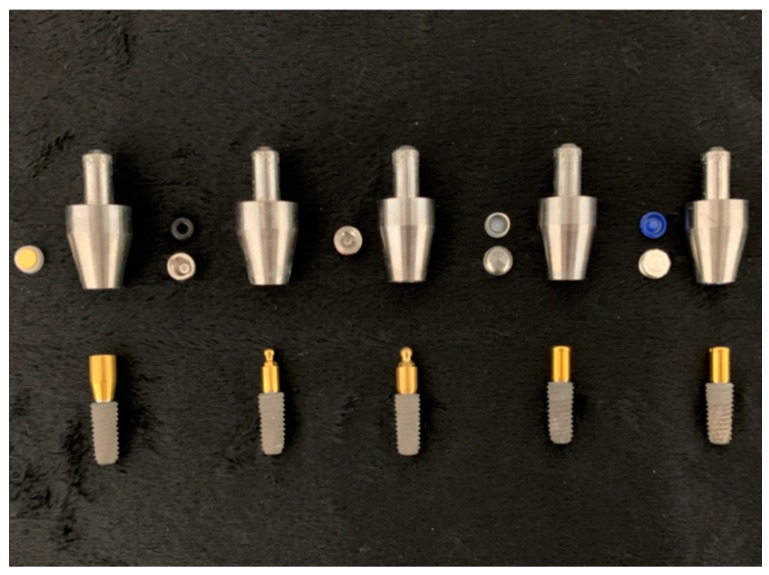
The five different implant attachment systems used in this study. Custom-made titanium-alloy jigs, attachment replacements, and implant fixture-tightened attachment abutments. From left to right: MAG, ORI, EZL, LOC, and KER.

**Figure 2 materials-12-03770-f002:**
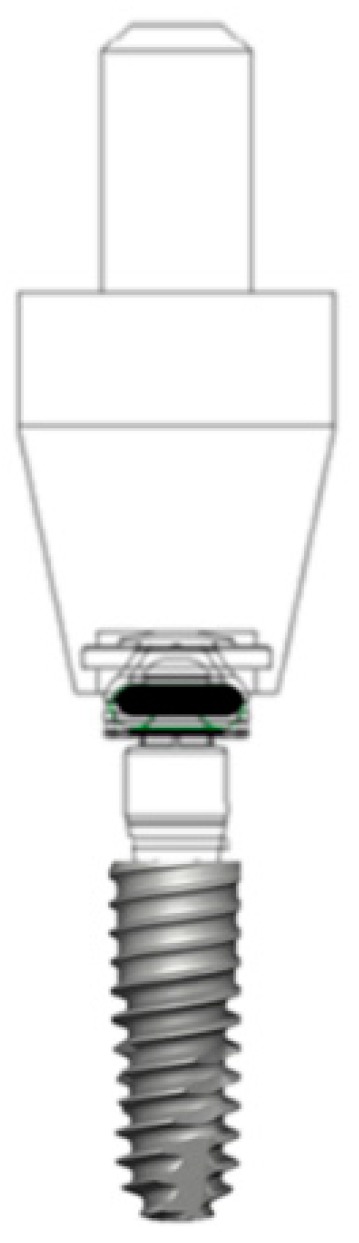
Schematic representation of the attachment systems.

**Figure 3 materials-12-03770-f003:**
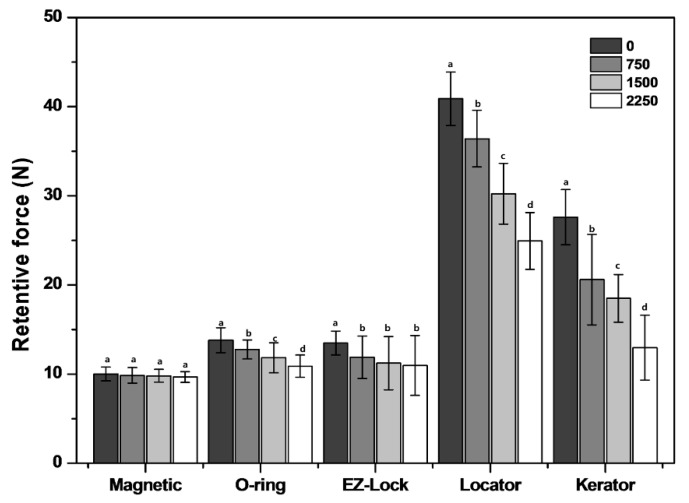
Changes in retentive forces observed in the insertion–removal cyclic test. Different letters above the bars indicate significant differences (*p* < 0.05) Error bars represent ± standard deviation of the mean.

**Figure 4 materials-12-03770-f004:**
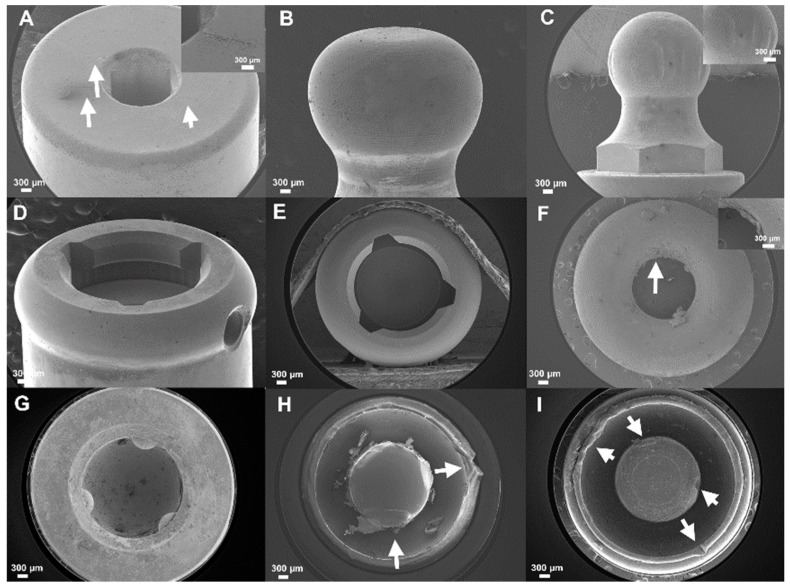
Field-emission scanning electron microscopy (FE-SEM) of the surface morphology of the attachments upon completion of 2250 insertion–removal cyclic loading. (**A**) MAG keeper, (**B**) ORI, (**C**) EZL, (**D**) LOC, (**E**) KER abutment, (**F**) ORI, (**G**) EZL, (**H**) LOC, and (**I**) KER replacement. Arrows indicate wear and deformed sites. Scale bar represents 300 µm.

**Table 1 materials-12-03770-t001:** The attachment systems investigated in this study.

Attachment Type	Abbreviation	Brand Name	Manufacturer	Materials of Replacements
Magnetic	**MAG**	Magfit^®^ SX-L	AICHI STEEL Co., Tokai, Japan	Magnet (NdFeB) with TiN coating
Ball	**ORI**	O-ring/TS Stud	OSSTEM IMPLAT Co., Seoul, Korea	Rubber
Ball	**EZL**	EZ Lock/SELA405SR	Samwon DMP Co., Yangsan, Korea	Ti-alloy ring and Zirconia ball (ZrO_2_)
Locator	**LOC**	Locator^®^/TS Port	HIOSSEN INC., Dallas, TX, USA	Nylon
Locator	**KER**	Kerator/Straight	DaeKwang IDM Co., Seoul, Korea	Nylon

**Table 2 materials-12-03770-t002:** Mean (N), Standard deviation (SD), and Coefficient of Variation (CV) of retentive forces observed in the insertion–removal cyclic test.

Cycle(s)	Mean ± SD (CV)				
	MAG	ORI	EZL	LOC	KER
**0 cycle**	10.01 ± 0.8 (0.08) ^Aa^	13.80 ± 1.4 (0.10) ^Ba^	13.49 ± 1.3 (0.10) ^Ba^	40.89 ± 3.0 (0.07) ^Da^	27.61 ± 3.1 (0.11) ^Ca^
**750 cycles**	9.85 ± 0.9 (0.09) ^Aa^	12.77 ± 1.1 (0.09) ^Bb^	11.89 ± 2.4 (0.20) ^Bb^	36.41 ± 3.2 (0.09) ^Db^	20.61 ± 5.1 (0.25) ^Cb^
**1500 cycles**	9.80 ± 0.7 (0.07) ^Aa^	11.83 ± 1.7 (0.14) ^Bc^	11.22 ± 3.0 (0.27) ^Bb^	30.23 ± 3.4 (0.11) ^Dc^	18.50 ± 2.7 (0.15) ^Cc^
**2250 cycles**	9.67 ± 0.6 (0.06) ^Aa^	10.89 ± 1.2 (0.11) ^Ad^	10.96 ± 3.4 (0.31) ^Ab^	24.95 ± 3.2 (0.13) ^Cd^	12.96 ± 3.6 (0.28) ^Bd^
**Average loss in retention (%) (0–2250 cycle)**	3.40%	21.09%	18.73%	38.98%	53.06%

Different uppercase superscript letters indicate statistically significant differences between attachment system groups (*p* < 0.05). Different lowercase superscript letters indicate statistically significant differences between cycles (*p* < 0.05).

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
