# Peer review of "In Vitro Effects of Cyclic Dislodgement on Retentive Properties of Various Titanium-Based Dental Implant Overdentures Attachment System"

_materials, 2019, doi:10.3390/ma12223770_

Round 1
Reviewer 1 Report
General remarks
The topic is interesting and important to warrant publication. The paper is written well, properly organized, and easy to follow. The information is presented in an open-minded and objective manner. However, the following minor concerns have to be revised before publication.
Minor revision:
Abstract- Please, add the No. of specimens for each group (n = 10). Introduction- The first sentence starts with the word- “recently”- overdentures on implants with attachment system were introduced long time ago and not recently. Please change the beginning of the sentence. Introduction- The Null Hypotheses are not clear- rewrite them Results- In “3.1. Retention force of specimens”- Line 4 starts with the number “3” –It is probably a mistake Results- In “3.2. Surface morphology”- Line 3 “(Fig. 4c)” should be- “(Fig. 4C)” M & M- You kept the testing solution and specimens at (37 ± 2) °C. Clinically, this temperature does not simulate oral environment since in the mouth there are different temperatures. Why you did not use thermo-cycling between water temperatures of 5° C and 55° C? An explanation is needed. In the discussion, you mentioned another study that used different temperatures and its influenced on the nylon of the attachment but there was no explanation why in the current study it was not used. Besides the explanation, it should be added in the study limitations as well. The order of all sections in the manuscript is not in accordance with the Journal specifications- ‘Materials & Methods’ should be after the ‘Discussion’ section. Please change accordingly Rewrite the references according to the Journal requirements. For example, Reference No 3. and 31.- the journal name should be after the title. Also, reference No. 25 and No. 36- There is no Journal name at all. Please, correct all the references. Please make a revision of the manuscript by English language editing.
Author Response
|
Reviewer #1 |
|
|
General Comments |
The topic is interesting and important to warrant publication. The paper is written well, properly organized, and easy to follow. The information is presented in an open-minded and objective manner. However, the following minor concerns have to be revised before publication. |
|
Response to General Comments |
The authors would like to thank the reviewer for the interest and in-depth review with valuable and helpful comments for improvement of the manuscript. |
|
|
|
|
Minor Concerns |
Abstract- Please, add the No. of specimens for each group (n = 10). |
|
Response to Minor Concerns1 |
We have now stated the number of specimens for each group (n = 10) in the abstract. |
|
|
|
|
Minor Concerns2 |
Introduction- The first sentence starts with the word- “recently”- overdentures on implants with attachment system were introduced long time ago and not recently. Please change the beginning of the sentence.
|
|
Response to Minor Concerns2 |
Thank you and we also agree that the use of IOD may not be that recent. We have deleted the word and revised the sentence.
|
|
|
|
|
Minor Concerns3 |
Introduction- The Null Hypotheses are not clear- rewrite them |
|
Response to Minor Concerns3 |
Thank you for your comments. We now have modified the null hypotheses that would be more suitable for this study. |
|
|
|
|
Minor Concerns 4 |
Results- In “3.1. Retention force of specimens”- Line 4 starts with the number “3” –It is probably a mistake Results- In “3.2. Surface morphology”- Line 3 “(Fig. 4c)” should be- “(Fig. 4C)” |
|
Response 4 |
Thank you for corrections. We have now revised mistyping and also corrected the figure numbering/alphabet to Fig. 4C. |
|
|
|
|
Minor Concerns 5 |
M & M- You kept the testing solution and specimens at (37 ± 2) °C. Clinically, this temperature does not simulate oral environment since in the mouth there are different temperatures. Why you did not use thermo-cycling between water temperatures of 5° C and 55° C? An explanation is needed. In the discussion, you mentioned another study that used different temperatures and its influenced on the nylon of the attachment but there was no explanation why in the current study it was not used. Besides the explanation, it should be added in the study limitations as well. |
|
Response 5 |
Thank you for comments. We agree that the thermo-cycling would provide more clinically relevant results. However, it would be difficult to have cyclic insertion-removal along with cyclic temperature changes. Hence, we considered technical environment of 37 °C. Still, as correctly suggested, this would be a limitation. Hence, now Discussion has been revised to add details about thermo-cycling (including some previous results) as well as stating the limitation of this study. |
|
|
|
|
Minor Concerns6 |
The order of all sections in the manuscript is not in accordance with the Journal specifications- ‘Materials & Methods’ should be after the ‘Discussion’ section. |
|
Response 6 |
Although we agree that some papers in Materials have been formatted in the way that reviewer has suggested, we have noted that this was not the case in the every journal. Also, we used the template provided by the journal which was in the order that is presented. We would like to have Materials and Methods first as it would be a logical sequence in terms of naming the attachment systems we used for this study followed by results of them. |
|
|
|
|
Minor Concerns 7 |
Please change accordingly Rewrite the references according to the Journal requirements. For example, Reference No 3. and 31.- the journal name should be after the title. Also, reference No. 25 and No. 36- There is no Journal name at all. Please, correct all the references. |
|
Response 7 |
All of references were now revisited and revised. Thank you for your comments. |
|
|
|
|
Minor Concerns 8 |
Please make a revision of the manuscript by English language editing. |
|
Response 8 |
The entire document has now been proofread by professional English editing service EDITAGE.
|
Reviewer 2 Report
Dear Authors, Your study is very interesting, overdenture is a really current topic in dentistry. And could be interesting to know which is the most predictable method.
In keyword section please use Medical subject Headings: https://meshb.nlm.nih.gov
Please subdivide introduction section into 2 subsection for a easy reading (background and aim)
In M&M section please specify degrees of cyclic insertion and disinsertion of the box on attachments. (0°?)
In discussion section please refer to bioengineering previous study on osstem implant too: 10.2174/1874210601812010219; 10.3390/biomedicines7010012; 10.3390/ma12111763
Please introduce an "abbreviation" section.
Thank You
Author Response
|
Reviewer #2 |
|
|
General Comments |
Your study is very interesting, overdenture is a really current topic in dentistry. And could be interesting to know which is the most predictable method. |
|
Response to General Comments |
We appreciate the valuable feedback and appreciation for the research work. |
|
|
|
|
Minor Concern1 |
Please subdivide introduction section into 2 subsection for a easy reading (background and aim) |
|
Response to Minor Concern1 |
Thank you for your comments. Now the introduction part has been amended as per the suggestions. |
|
|
|
|
Minor Concerns 2 |
In keyword section please use Medical subject Headings: https://meshb.nlm.nih.gov
|
|
Response 2 |
Keywords are now in accordance to Medical subject Headings |
|
|
|
|
Minor Concerns3 |
In M&M section please specify degrees of cyclic insertion and disinsertion of the box on attachments.
|
|
Response3 |
We now have added specific information throughout the document. |
|
|
|
|
Minor Concerns 4 |
In discussion section please refer to bioengineering previous study on osstem implant too: 10.2174/1874210601812010219; 10.3390/biomedicines7010012; 10.3390/ma12111763 |
|
Response 4 |
As per the suggestions, the discussion section has been modified to include reference on view of the two articles suggested. |
|
|
|
|
Minor Concerns 5 |
Please introduce an "abbreviation" section. |
|
Response 5 |
As suggested by the reviewer, we have included the abbreviation section in the Table 1. |
|
|
|
Round 2
Reviewer 2 Report
Dear Authors, thank You for Your prompt reply and for agreeing my suggestions.
Now the manuscript is suitable for publication.
Kind Regards